# MAPKs Are Highly Abundant but Do Not Contribute to α_1_-Adrenergic Contraction of Rat Saphenous Arteries in the Early Postnatal Period

**DOI:** 10.3390/ijms22116037

**Published:** 2021-06-03

**Authors:** Dina K. Gaynullina, Tatiana V. Kudryashova, Alexander V. Vorotnikov, Rudolf Schubert, Olga S. Tarasova

**Affiliations:** 1Department of Human and Animal Physiology, Faculty of Biology, M.V. Lomonosov Moscow State University, 119234 Moscow, Russia; ost.msu@gmail.com; 2Department of Physiology, Russian National Research Medical University, 117513 Moscow, Russia; 3Department of Internal Medicine, Division of Pulmonary, Critical Care and Sleep Medicine, University of California Davis, Davis, CA 95616, USA; tkud@ucdavis.edu; 4National Medical Research Center of Cardiology, Institute of Experimental Cardiology, 121552 Moscow, Russia; a.vorotnikov@cardio.ru; 5Physiology, Institute of Theoretical Medicine, Medical Faculty, University of Augsburg, 86159 Augsburg, Germany; rudolf.schubert@med.uni-augsburg.de; 6Laboratory of Exercise Physiology, State Research Center of the Russian Federation-Institute for Biomedical Problems, Russian Academy of Sciences, 123007 Moscow, Russia

**Keywords:** p38 MAPK, p42/44 MAPK, ERK1/, early postnatal ontogenesis, smooth muscle, functional remodeling

## Abstract

Previously, the abundance of p42/44 and p38 MAPK proteins had been shown to be higher in arteries of 1- to 2-week-old compared to 2- to 3-month-old rats. However, the role of MAPKs in vascular tone regulation in early ontogenesis remains largely unexplored. We tested the hypothesis that the contribution of p42/44 and p38 MAPKs to the contraction of peripheral arteries is higher in the early postnatal period compared to adulthood. Saphenous arteries of 1- to 2-week-old and 2- to 3-month-old rats were studied using wire myography and western blotting. The α_1_-adrenoceptor agonist methoxamine did not increase the phosphorylation level of p38 MAPK in either 1- to 2-week-old or 2- to 3-month-old rats. Accordingly, inhibition of p38 MAPK did not affect arterial contraction to methoxamine in either age group. Methoxamine increased the phosphorylation level of p42/44 MAPKs in arteries of 2- to 3-month-old and of p44 MAPK in 1- to 2-week-old rats. Inhibition of p42/44 MAPKs reduced methoxamine-induced contractions in arteries of 2- to 3-month-old, but not 1- to 2-week-old rats. Thus, despite a high abundance in arterial tissue, p38 and p42/44 MAPKs do not regulate contraction of the saphenous artery in the early postnatal period. However, p42/44 MAPK activity contributes to arterial contractions in adult rats.

## 1. Introduction

Early postnatal maturation is associated with growth and development of different organs and systems, including the circulatory system. The development of the vascular system is a complex process and occurs at different structural and functional levels, from the systemic (maturation is accompanied by an increase in blood pressure [1,2]) to the molecular (maturational changes in protein expression and of the mechanisms of vascular smooth muscle contractility control [3,4,5,6]). Functional remodeling of vascular smooth muscle during postnatal development includes (but is not limited to) an increase in the Ca^2+^-dependent control of contractility and a decrease in the Ca^2+^-sensitivity of the contractile apparatus [3].

MAPKs were shown to regulate the Ca^2+^-sensitivity of vascular smooth muscle contraction, due to augmenting MLCK activity [7] and also due to a PKC-dependent weakening of the inhibitory effect of caldesmon [8,9]. Therefore, both p38 and p42/44 MAPKs increase smooth muscle cell contractility in response to a variety of stimuli [8,10,11,12,13,14,15,16,17]. In particular, the contribution of p42/44 and p38 MAPKs to vascular contraction to the α_1_-adrenoceptor agonist phenylephrine has been shown previously in the abdominal aorta of adult rats [17]. Of note, α_1_-adrenoceptors are very important for the regulation of vascular tone, especially in small resistance arteries, where they mediate the vasoconstrictor influence of sympathetic nerves. In addition, MAPKs activation, detected by their site-specific phosphorylation, may differ in time dynamics from the vasocontractile response, which suggests their participation in other parallel processes, besides smooth muscle contraction [18].

Our previous study has revealed that the abundance of p42/44 and p38 MAPK proteins is considerably higher in muscular type saphenous arteries of 1- to 2-week-old rats in comparison to adult ones [3]. Similarly, in elastic type arteries (ovine carotid artery), p42/44 MAPKs were shown to be more abundant in near-term fetuses in comparison to adult ewes [19]. Further, more pronounced effects of p42/44 MAPK inhibitors on contractile responses have been shown in ovine carotid artery of fetuses compared to adult ewes [19], suggesting a role of MAPKs in the remodeling of contractility mechanisms during the early postnatal period. Indeed, activation of α_1_-adrenoceptors led to phosphorylation of p42/44 MAPKs in cerebral arteries of fetal ewes, but not in those of adult ewes [20,21]. All these data point to a greater functional role of p42/44 MAPKs in the regulation of contraction, at least in relatively large elastic type arteries in early ontogenesis. However, the role of MAPKs, particularly p38, in the regulation of contraction in immature small muscular type arteries from peripheral circulation is relatively unexplored. Thus, we tested the hypothesis that the contribution of p42/44 and p38 MAPKs to α_1_-adrenergic contraction of peripheral muscular type arteries is higher in rats during the early postnatal period, compared to adults.

## 2. Results

First, we compared the protein content of p38 and p42/44 MAPKs in saphenous arteries of 1- to 2-week-old and 2- to 3-month-old rats. As before [3], the content of p38 and p42/44 MAPKs was considerably higher in arteries of 1- to 2-week-old rats in comparison to 2- to 3-month-old rats (Figure 1a–c).

Further, we evaluated whether p38 and p42/44 MAPKs are activated during α_1_-adrenoceptor stimulation in saphenous arteries of 2- to 3-month-old and 1- to 2-week-old rats by measuring their phosphorylation level upon activation by a submaximal concentration of the α_1_-adrenoceptor agonist methoxamine (MX, up to 10 µM, see Methods). In addition, the effects of MAPK inhibitors on arterial contractile responses to MX were also studied.

We did not detect differences in the phosphorylation level of p38 MAPK before and during MX-induced stimulation in arteries of either 2- to 3-month-old (Figure 2a) or 1- to 2-week-old (Figure 2b) rats. Moreover, we did not observe effects of two different p38 MAPK inhibitors, SB202190 [22] and SB220025 [23] on the contractile response to MX in arteries of both age groups (Figure 2c,d). Therefore, despite a higher content of p38 MAPK in arteries of 1- to 2-week-old rats, the data suggest that it does not regulate their contractile responses to MX.

Further, we compared phosphorylation levels of p42/44 MAPKs before and during the MX-induced stimulation of arteries of 2- to 3-month-old and 1- to 2-week-old rats and the effects of a p42/44 MAPK inhibitor on MX-induced contractile responses (Figure 3a–d). The phosphorylation level of p42 MAPK increased after MX-induced stimulation in arteries of 2- to 3-month-old rats; this was not observed in arteries of 1- to 2-week-old rats. The phosphorylation level of p44 MAPK after MX-induced stimulation increased slightly (approximately 1.5 times, Figure 3d) in arteries of 1- to 2-week-old rats, and was considerably augmented in arteries of 2- to 3-month-old rats (approximately 3 times, Figure 3c). The higher degree of p42/44 MAPK activation in arteries of 2- to 3-month-old animals correlated well with the effects of the p42/44 MAPK inhibitor U0126 [22]. In comparison to its inactive analogue U0124, U0126 decreased contractile response to MX in arteries of 2- to 3-month-old animals (Figure 3e); this was not observed in arteries of 1- to 2-week-old rats (Figure 3f). These data show that p42/44 MAPKs are important for fully manifested contractions of arteries in 2- to 3-month-old rats, but not in those 1- to 2-weeks old.

## 3. Discussion

### 3.1. p38 MAPK Does Not Regulate α_1_-Adrenergic Contractions of Saphenous Arteries at Both Ages

Two sets of our results show that p38 MAPK does not regulate α_1_-adrenoceptor-mediated contractions of the muscular type saphenous artery of either adult or 1- to 2-week-old rats. First, we did not observe an increase in the p38 phosphorylation level in response to MX in any age group, i.e., p38 MAPK is not activated upon α_1_-adrenergic stimulation in arteries of both age groups.

Second, we studied the effects of two structurally different p38 MAPK inhibitors, SB202190 and SB220025 [22,23], on arterial contractile responses in two age groups. The concentration of the p38 MAPK inhibitors used was selected based on our results of a separate experimental series involving a higher concentration of SB202190. In parallel, SB202474, an inactive analogue of SB202190, was used as a control for non-specific effects of the inhibitor. In saphenous arteries of 1- to 2-week-old rats, both SB202190 (10 µM) and SB202474 considerably reduced contractile responses to MX compared to vehicle-treated preparations (Appendix A). Thus, higher concentrations of the p38 MAPK inhibitor cannot be used, since they show non-specific effects, at least in the immature vasculature of younger rats. Therefore, an inhibitor concentration of 1 µM was chosen for the main experimental series. In this concentration, both SB202190 and SB220025 had no effect on arterial contractile responses of either 2- to 3-month-old or 1- to 2-week-old rats.

Notably, a contribution of p38 MAPK to α_1_-adrenergic contraction of the aorta was previously described for adult rats [17]. In addition, p38 MAPK was shown to participate in the norepinephrine-induced actin polymerization and contraction of rat small mesenteric arteries [15]. Moreover, in several other studies p38 MAPK was shown to regulate the contraction of rat aorta to AngII [24], ET-1 [8,11] and serotonin [8], the contraction of rat common carotid arteries to serotonin [12], as well as pressure-induced contraction of rat skeletal muscle arterioles [25]. However, some studies do not support the idea of p38 MAPK participation in the control of adrenergic contractions of arteries. For example, no role of p38 MAPK was found in the contraction of the aorta to phenylephrine [24] or skeletal muscle arterioles to norepinephrine [25]. Altogether these data point to a vascular bed- and agonist-specific activation of p38 MAPK and, consequently, role in the contractile responses of the adult vasculature. To the best of our knowledge, the of role p38 MAPK in the saphenous artery has never been studied before. We cannot exclude that p38 MAPK regulates non-adrenergic contractions of saphenous arteries, however this is beyond the topic of the present study.

Developmental alterations in the functional activity of different proteins regulating smooth muscle contraction have been shown previously by our and other groups [3,5,6,26]. However, to the best of our knowledge, the developmental alteration in the functional role of p38 MAPK has never been studied before. Our novel data suggest that, despite a high abundance of p38 MAPK in saphenous arteries of 1- to 2-week-old rats, it does not contribute to α_1_-adrenoceptor mediated contractions. Similar to the adult vasculature, the role of p38 MAPK in non-adrenergic contractions of younger rats is possible, but needs to be addressed in future studies.

### 3.2. p42/44 MAPK Regulates α_1_-Adrenergic Contractile Responses of Saphenous Arteries of 2- to 3-Month-Old, but Not 1- to 2-Week-Old Rats

Our data show that p42/44 MAPKs are phosphorylated and thereby activated upon α_1_-adrenoceptor stimulation in saphenous arteries of adult rats. Moreover, inhibition of p42/44 MAPK reduced α_1_-adrenergic contractions. These data are in accordance with previously published data from rat aorta where contractions to an α_1_-adrenergic agonist were shown to be partially mediated by p42/44 MAPK [17]. Further, a procontractile role of p42/44 MAPK was reported for ET-1-induced contractions of rat aorta [8,11], 5HT-induced contractions of rat carotid arteries [12,13] as well as for ANG II-induced contractions of skeletal muscle arterioles [25].

Developmental alterations in the functional contribution of p42/44 MAPK to arterial contraction were studied previously for ovine carotid [19] and cerebral arteries [20], but not for muscular type arteries of the systemic circulation, such as the saphenous artery. Activation of α_1_-adrenoceptors increased the phosphorylation level of p42/44 MAPK in fetal, but not in adult, ewe middle cerebral arteries [20,21]. A higher content of p42/44 MAPK was observed in fetal compared to adult ewe carotid arteries [19]. This was accompanied by an elevated procontractile role of p42/44 MAPK in carotid arteries of fetal ewes [19].

In contrast to developing cerebral arteries, in the saphenous artery, i.e., a muscular type artery of the peripheral circulation, the activation of p42/44 MAPK upon stimulation of α_1_-adrenoceptors occurs predominantly in adult, but less so in 1- to 2-week-old rats (where we observed weak phosphorylation of p44, but no phosphorylation of p42 MAPK). Thus, the relatively high abundance of p42/44 MAPK proteins in younger arteries does not play a role in α_1_-adrenergic contractions in 1- to 2-week-old rats. Similar to p38 MAPK, the functional contribution of p42/44 MAPK to non-adrenergic contractions of developing arteries is the topic of future studies.

## 4. Materials and Methods

### 4.1. Animals

Experiments were performed in accordance with the European Convention on the protection of animals used for scientific purposes (EU Directive 2010/63/EU). Animal procedures were approved by the Biomedical Ethics Committee of the Russian Federation State Research Center-Institute for Biomedical Problems, Russian Academy of Sciences (protocol 426, approval date 20 June 2016). We used 10- to 15-day-old (referred as 1- to 2-week-old rats) and 2.5- to 3.5-month-old (i.e., 2- to 3-month-old rats) male Wistar rats. Rats were sacrificed under CO_2_ anesthesia by decapitation. Saphenous arteries were used as representatives for peripheral muscular type arteries.

### 4.2. Wire Myography

Saphenous arteries were isolated in a solution for vessel isolation (for composition see below) and thereafter mounted in an isometric myograph (DMT, Denmark, model 610 M). The endothelium was denuded using a rat whisker. All vessels were kept at 37 °C in a solution for myograph experiments (for composition see below) and bubbled with a gas mixture (95% O_2_, 5% CO_2_). Force transducer signals were digitalized at 1 kHz and recorded on a PC hard drive using the PowerLab 4/30 system (ADInstruments, Colorado Springs, CO, USA) and LabChart software (ADInstruments, USA). Vessel preparations were stretched radially to a lumen diameter corresponding to 0.9d_100_; where d_100_ is the inner arterial diameter of relaxed arteries subjected to a transmural pressure of 100 mmHg [27]. Thereafter, preparations were activated twice with 10 μM phenylephrine and successful endothelium removal was checked subsequently using 10 μM acetylcholine during 1–3 μM phenylephrine-induced precontraction.

Contractile responses were studied during cumulative addition of the α_1_-adrenoceptor agonist methoxamine (MX, in the concentration range from 0.01 µM to 100 µM). During each experiment, two concentration-response relationships to MX were obtained. The first was used to ensure comparable initial sensitivity of the vessels further treated with either an inhibitor or its solvent. The second was used to study the effects of different inhibitors in comparison to the application of the same volume of solvent (DMSO) or the same concentration of an inactive inhibitor analogue. Drugs or DMSO were added 25 min before the second MX concentration-response relationship.

Active force values were calculated by subtraction of the passive force values recorded from the preparation with fully relaxed smooth muscle (at the beginning of each experiment in the presence of 1 µM of the NO-donor DEA/NO). All active force values of the second concentration-response relationship were expressed as the percentage of the maximum active force achieved during the first concentration-response relationship. The second concentration-response relationships are shown in the figures.

### 4.3. Western Blotting

For one preparation, three 2-mm segments of saphenous arteries of 2- to 3-month-old rats, or nine 2-mm segments of saphenous arteries of 1- to 2-week-old rats, were used. The segments were first separately mounted in a wire myograph (model 410A), normalized, then twice activated with 10 μM phenylephrine in the same way as described above for myograph experiments. Some preparations were twice activated with 10 μM phenylephrine and 30 min later instantly frozen in liquid nitrogen (non-activated samples). Other preparations, after activation with 10 μM phenylephrine (twice), were treated by cumulative addition of MX (in the range of 0.1 µM to 10 µM) similar to concentration-response relationships in myograph experiments. After the last concentration of MX, the vessel segment was instantly frozen in liquid nitrogen (activated samples). Thereafter, all samples were homogenized in SDS-buffer (for composition see below), boiled for 5 min, centrifuged at 12,000 rpm for 5 min and the supernatant was collected for further analysis.

Proteins were separated by SDS PAGE and transferred onto polyvinylydenedifluoride membranes. The membranes were blocked in 5% nonfat milk in Tris-Buffered Saline supplemented with Tween 20 (TBST, for composition see below) and sequentially incubated with antibodies against p38 MAPK (cell signaling, 1:1000 in TBSt with 1% milk) or phospho-p38 MAPK-Thr180/Tyr182 (cell signaling, 1:1000 in TBSt with 1% milk) or p42/44 MAPK (cell signaling, 1:1000 in TBSt with 1% milk) or phospho-p42/44 MAPK- Thr202/Tyr204 (cell signaling, 1:1000 in TBSt with 1% milk) or GAPDH (Abcam, 1:2000 in TBSt with 1% milk). Thereafter, membranes were incubated with appropriate secondary antibodies (antirabbit (GE Healthcare, 1:5000) or antimouse (Amersham, 1:10,000)). The protein bands were visualized by an Enhanced Chemiluminescence (ECL) protocol using the SuperSignal West Dura Substrate (Pierce, ThermoFisher Scientific, Waltham, MA, USA). To ensure linearity of the chemiluminescence signal, the membranes were exposed to the film for various time periods, then scanned using a GS-800 densitometer (Bio-Rad, Heracles, CA, USA), and quantitatively processed with the use of the Quantity One software (Bio-Rad, Heracles, CA, USA). Protein content was estimated by the optical density of the band, multiplied by its area. Background intensity was determined in 3 different regions on the film. This was used to correct the relevant bands for the background.

The data on total protein content were normalized relative to GAPDH and the average value of the protein/GAPDH ratios in the 2- to 3-month-old group was taken as 100%. The data on phospho-p38 MAPK and phospho-p42/44 MAPK were normalized to respective values for total p38 MAPK and p42/44 MAPK. Considering the data on isoform-specific roles of p42 and p44 in regulation of cellular processes [28], we estimated their phosphorylation levels separately. Obtained values were expressed as a percentage of the mean value in the appropriate non-activated group.

### 4.4. Solutions

(1)Solution for vessel isolation, in mM: 145 NaCl, 4.5 KCl, 1.2 NaH_2_PO_4_, 1 MgSO_4_, 0.1 CaCl_2_, 0.025 EDTA and 5 HEPES.(2)Solution for myograph experiments, in mM: 120 NaCl, 26 NaHCO_3_, 4.5 KCl, 1.2 NaH_2_PO_4_, 1.0 MgSO_4_, 1.6 CaCl_2_, 5.5 glucose, 0.025 EDTA and 5 HEPES; equilibrated with 5% CO_2_ in 95% O_2_.(3)SDS-buffer: 10% water-free glycerin, 1% SDS, 0.0625 mM tris-HCl, 5% β-mercaptoethanol and 0.05% bromphenol blue.(4)TBST: 20 mM tris-HCl, 20 mM NaCl, 0.1% Tween 20, pH 7.6.

### 4.5. Statistical Analysis

All data are presented as mean ± SEM; n represents the number of animals or the number of samples in western blotting experiments. Statistical significance was determined using Repeated Measures ANOVA or the unpaired Student *t*-test, as appropriate. Statistical significance was reached at *p* < 0.05.

## 5. Conclusions

To conclude, our data demonstrate that, despite a high abundance of p38 and p42/44 MAPKs in peripheral muscular type arteries of rats in early postnatal ontogenesis, they do not participate in the regulation of their contractile responses to an agonist of α_1_-adrenoceptors. In this respect, peripheral arteries differ from cerebral arteries from perinatal animals, in which the development of contractile responses can be partially associated with activation of p42/44 MAPKs. However, p42/44 MAPK is important for fully manifested arterial contractions in adult rats. Thus, arterial functional remodeling during early postnatal ontogenesis is not associated with a decrease in the role of p38 and p42/44 

## Figures and Tables

**Figure 1 ijms-22-06037-f001:**
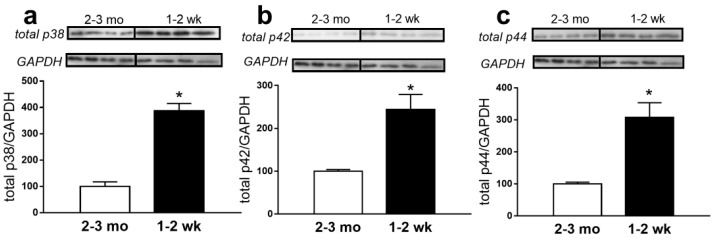
p38, p42 and p44 MAPK protein abundance is higher in saphenous arteries of 1- to 2-week-old rats. Protein abundance of MAPK p38 (**a**), *n* = 4, p42 (**b**), *n* = 4 and p44 (**c**), *n* = 4 in saphenous arteries of 2- to 3-month-old and 1- to 2-week-old rats. * *p* < 0.05 (unpaired Student *t*-test).

**Figure 2 ijms-22-06037-f002:**
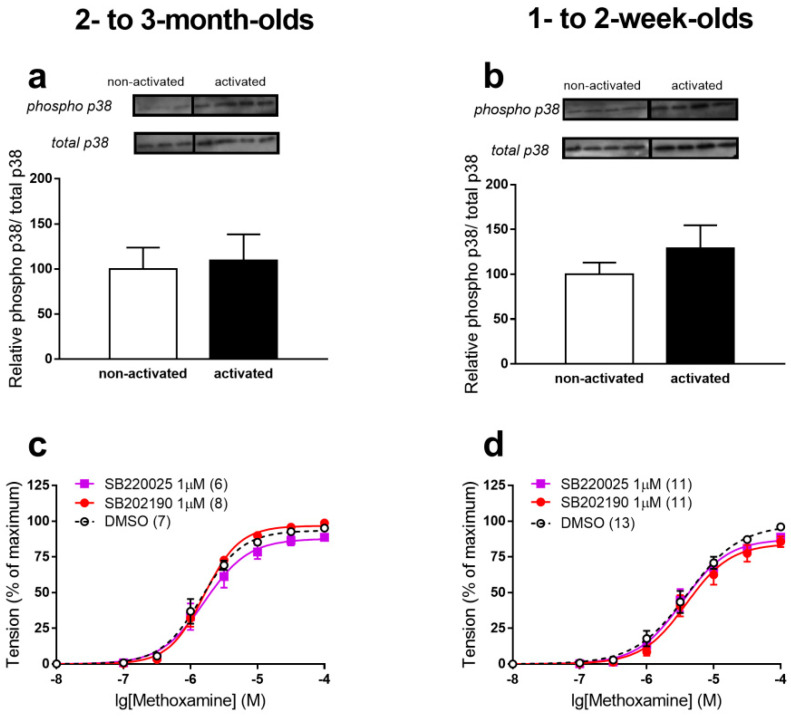
p38 MAPK does not contribute to contractile responses of rat saphenous arteries to MX. (**a**,**b**). Phosphorylation level of p38 MAPK in arteries of 2- to 3-month-old ((**a**), *n* = 3 for non-activated and *n* = 4 for activated vessels) and 1- to 2-week-old ((**b**), *n* = 4 for non-activated and *n* = 4 for activated vessels) rats. (**c**,**d**). Concentration-response relationships to MX in the presence of 1 µM SB202190, 1 µM SB220025 or DMSO (vehicle of the inhibitors) of arteries from 2- to 3-month-old (**c**) and 1- to 2-week-old (**d**) rats. Numbers in parenthesis indicate the number of animals.

**Figure 3 ijms-22-06037-f003:**
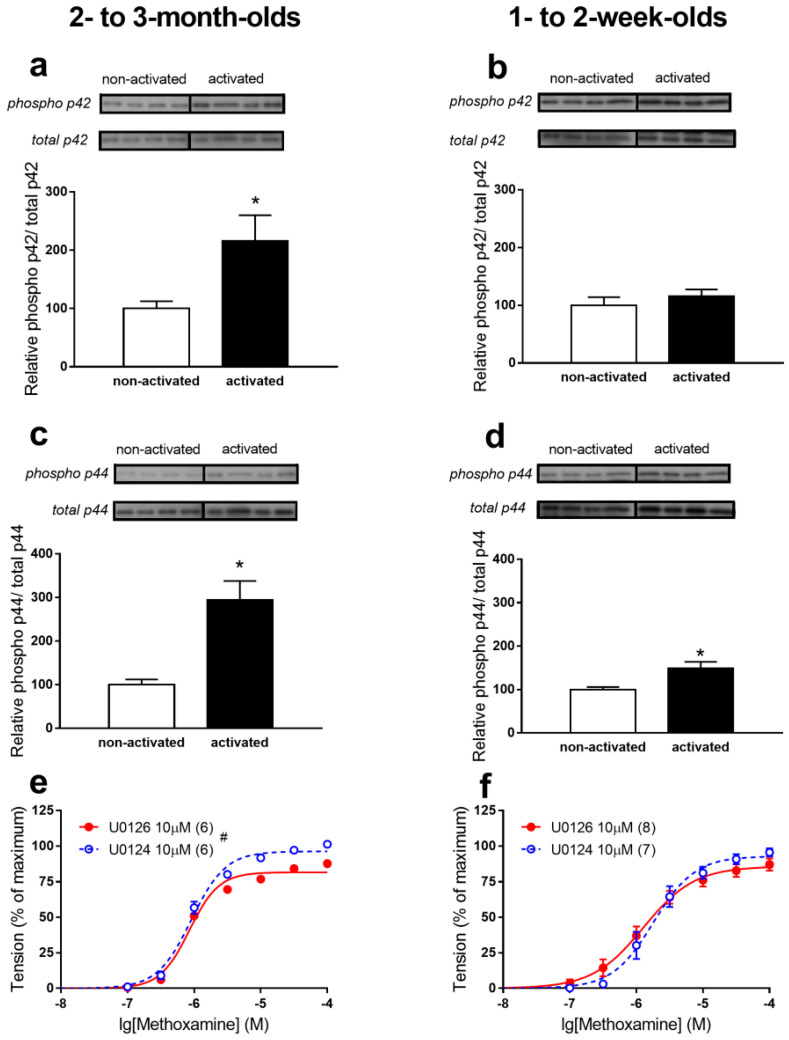
p42/44 MAPKs regulate contractile responses of saphenous arteries in 2- to 3-month-old, but not 1- to 2-week-old rats. (**a**–**b**). Phosphorylation level of p42 MAPK in arteries of 2- to 3-month-old ((**a**), *n* = 4 for non-activated and *n* = 4 for activated vessels) and 1- to 2-week-old ((**b**), *n* = 4 for non-activated and *n* = 4 for activated vessels) rats. (**c-d**). Phosphorylation level of p44 MAPK in arteries of 2- to 3-month-old ((**c**), *n* = 4 for non-activated and *n* = 4 for activated vessels) and 1- to 2-week-old ((**d**), *n* = 4 for non-activated and *n* = 4 for activated vessels) rats. (**e**–**f**). Concentration-response relationships to MX in the presence of 10 µM U0126 or 10 µM U0124 of arteries from 2- to 3-month-old (**e**) and 1- to 2-week-old (**f**) rats. * *p* < 0.05 (unpaired Student *t*-test), # *p* < 0.05 (Repeated Measures ANOVA). Numbers in parenthesis indicate the number of animals.

## Data Availability

All data generated during this study are available from the corresponding author on reasonable request.

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
