# Peer review of "MAPKs Are Highly Abundant but Do Not Contribute to α1-Adrenergic Contraction of Rat Saphenous Arteries in the Early Postnatal Period"

_ijms, 2021, doi:10.3390/ijms22116037_

Round 1

Reviewer 1 Report

The manuscript authored by Gaynullina DK, et al. reports that MAPKs are not relevant to contraction of alpha1 activation in the early postnatal period. 

Major concerns:

  1. Introduction: The significance of MAPKs in muscle contractility of alpha-1 receptor was not addressed. There was insufficient description on alpha-1 receptor in this section.
  2. Results: Figure 1: In comparison with 2-3 months, the background of western blots for protein expression of p38, p42, and p44 was too dark to delineate the difference, esp. for b) and c). The significant results of p42 and p44 are questionable.
  3. Method: The methods to quantify expression of proteins must be provided.   

Author Response

The manuscript authored by Gaynullina DK, et al. reports that MAPKs are not relevant to contraction of alpha1 activation in the early postnatal period. 

We wish to thank the Reviewer for careful evaluation of our manuscript. To meet the points raised by the reviewer we modified the manuscript. The changes in the manuscript text are marked in red.

Major concerns:

  1. Introduction: The significance of MAPKs in muscle contractility of alpha-1 receptor was not addressed. There was insufficient description on alpha-1 receptor in this section.

Our response:

In accordance with the suggestion of the Reviewer, the following sentences were added to the Introduction section, describing the significance of MAPKs to vascular contraction to α1 –adrenoceptors (Page 2):

“In particular, the contribution of p42/44 and p38 MAPKs to vascular contraction to the α1-adrenoceptor agonist phenylephrine has been shown previously in the abdominal aorta of adult rats. Of note, α1-adrenoceptors are very important for the regulation of vascular tone, especially in small resistance arteries, where they mediate the vasoconstrictor influence of sympathetic nerves.” (Akinaga et al., 2019 doi: 10.1111/bph.14617; Anschutz, Schubert, 2005 doi: 10.1038/sj.bjp.0706323).

  1. Results: Figure 1: In comparison with 2-3 months, the background of western blots for protein expression of p38, p42, and p44 was too dark to delineate the difference, esp. for b) and c). The significant results of p42 and p44 are questionable.

Our response:

We agree with the Reviewer that the background intensity of the original blots is different between samples of 1- to 2-week-old and 2- to 3-month old rats. Importantly, this was accounted for by determining the background intensity in 3 different regions of the film; this was used to correct the relevant bands for the background. This information is given in the methods section (page 8). Of note, the results of the present study are supported by similar results in our previous study where quantitatively comparable differences in protein abundance of p38 and p42/44 (ERK1/2) MAPKs between 1- to 2-week-old and 2- to 3-month old rats were shown (Puzdrova et al., 2014, doi: 10.1111/apha.1233). Based on the detailed explanation above, we don’t think that our significant results are questionable. Moreover, the effect size, i.e. considerable difference in MAPK protein abundance between of 1- to 2-week-old and 2- to 3-month old rats strongly suggests a relevant difference.

  1. Method: The methods to quantify expression of proteins must be provided.   

Our response:

The following information was added to the text of the Methods section (page 8):

“Protein content was estimated by the optical density of the band, multiplied by its area. Background intensity was determined in 3 different regions on the film. This was used to correct the relevant bands for the background.”

Reviewer 2 Report

Gaynullina et al investigated the contribution of p42/44 and p38 MAPKs to α1-adrenergic contraction of peripheral muscular type arteries is higher in rats during the early postnatal period compared to adults. The manuscript is seemed well written. I'd like to congratulate you to chose this important topic. I have some small points. Please see my comments below;

*You could give some results regarding levels of MAPK proteins.
*Maybe you could exchange the box plots and graphs (you can put the results of younger on the left instead) for all figures.
*You do not need to give statistical methods as a footnote of figures

Author Response

Gaynullina et al investigated the contribution of p42/44 and p38 MAPKs to α1-adrenergic contraction of peripheral muscular type arteries is higher in rats during the early postnatal period compared to adults. The manuscript is seemed well written. I'd like to congratulate you to chose this important topic. I have some small points. Please see my comments below;

We wish to thank the Reviewer for careful evaluation of our manuscript. To meet the points raised by the reviewer we modified the manuscript. The changes in the manuscript text are marked in red.

*You could give some results regarding levels of MAPK proteins.

Our response: Protein abundance of p42/44 and p38 MAPKs was considerably higher in arteries of 1- to-2-week-old rats compared to 2- to 3-month-old animals (Fig. 1 of the present manuscript and Fig. 8 in the paper by Puzdrova et al., 2014 doi: 10.1111/apha.1233). All MAPK bands were well detected by antibodies, however, we did not directly compare the content of different isoforms in the same age group (considering potentially different affinity of antibodies to different MAPK isoforms). In addition, we estimated phosphorylation levels of p42 and p44 separately considering the data on their isoform-specific roles in regulation of cellular processes (doi 10.3390/cells9010038) (this information was added to the text, page 8).

*Maybe you could exchange the box plots and graphs (you can put the results of younger on the left instead) for all figures.

Our response: With due respect to the opinion of the Reviewer, we would prefer to leave the order of data presentation as it is now. The point is that we are comparing 1- to 2-week-old animals with 2- to 3-month-old rats. Since the 2- to 3-month-old animals are the "reference point", they are in the first (left) place in the figures.

*You do not need to give statistical methods as a footnote of figures

Our response: We respectfully disagree with the reviewer. In order to facilitate transparency for the reader of the statistical methods used we would like to leave the methods in the legends.

Round 2

Reviewer 1 Report

Some concerns have been addressed appropriately in the revised manuscript. The major issue still stands.

Major: 

The major concern was the results of Western blotting for p42 and p44, especially Fig. 1. As authors explained in the letter, the intensity of protein expression was calculated on the basis of background. This explanation is not solid. It is very difficult to be convinced that the expression of p42 in 1-2 weeks was more than two-fold than 3-4 months and the expression of p44 in 1-2 weeks was more than three-fold than 3-4 months. These results play major role in conclusion of this manuscript.    

Author Response

Our response:

We would like to thank the Reviewer for the careful evaluation of our manuscript that helped us to improve it.

In accordance with the comment of the Reviewer asking for a more convincing presentation of the data in Fig. 1, we present now more representative examples of the Western Blots with a lighter and more uniform background. Accordingly, the average data were recalculated. Importantly, they are very similar to the previous data.

Round 3

Reviewer 1 Report

The major concerns have been addressed appropriately in the revised manuscript.